# Effect of Different Etching Times with Hydrofluoric Acid on the Bond Strength of CAD/CAM Ceramic Material

**DOI:** 10.3390/ma15207071

**Published:** 2022-10-11

**Authors:** Liane Tabitha Avram, Sergiu-Valentin Galațanu, Carmen Opriș, Cristian Pop, Anca Jivănescu

**Affiliations:** 1Department of Prosthodontics, Faculty of Dental Medicine, University of Medicine and Pharmacy “Victor Babeș” Timișoara, Revoluţiei din 1989 Bd. No. 9, 300070 Timișoara, Romania; 2Department of Mechanics and Strength of Materials, Politehnica University of Timisoara, Mihai Viteazu Bd., 300222 Timisoara, Romania; 3Department of Materials and Manufacturing Engineering, Politehnica University of Timisoara, Mihai Viteazu Bd., Timis, 300222 Timisoara, Romania; 4Department of Mechatronics, Politehnica University of Timisoara, Mihai Viteazu Bd., 300222 Timisoara, Romania; 5TADERP Research Center, University of Medicine and Pharmacy “Victor Babeș” Timișoara, Revoluţiei din 1989 Bd. No. 9, 300070 Timișoara, Romania

**Keywords:** CAD/CAM, micro-shear bond strength, glass ceramic, hydrofluoric acid, etching time, surface treatment, lithium disilicate, leucite-reinforced ceramic, hybrid ceramic

## Abstract

The objective of this study was to evaluate the influence of hydrofluoric acid (HF) and conditioning time on the micro-shear bond strength (µSBS) between dual-cure resin cement and glass-ceramic materials, such as lithium disilicate ceramic (IPS e.max CAD, Ivoclar Vivadent) (EX) and leucite-reinforced ceramic (IPS Empress CAD, Ivoclar Vivadent) (EP), and also a hybrid ceramic (Vita Enamic, Vita Zahnfabrik) (VE). Eighteen sections with 1 mm thickness were cut from each CAD/CAM material and randomly divided into three groups, according to the surface etching time (30 s, 60 s, 90 s). The surface treatment was performed using 9.5% HF acid gel, then resin cement was applied on the prepared ceramic plates and light cured. µSBS values between resin cement and the ceramic material were measured with a universal testing machine at a crosshead speed of 0.5 mm/min until the failure occurred. The fractured surfaces of specimens were microscopically evaluated, and failure modes were classified as: adhesive between resin cement and ceramic, cohesive within ceramic or cement and mixed failure. Surface roughness of etched samples was examined using a scanning electron microscope. Obtained data were statistically analysed using one-way analysis of variance (ANOVA) and Bonferroni post hoc test with a level of significance α = 0.05. The results of the statistical methods applied indicate that µSBS mean difference for leucite-reinforced ceramic (EP) was statistically significant (*p* < 0.05). However, µSBS values for hybrid ceramic (VE) and lithium disilicate ceramic (EX) were not affected, from a statistical point of view, by the conditioning time (*p* > 0.05).

## 1. Introduction

The vast diversity of the restorative materials has increased the use of chairside CAD/CAM restorations in the last ten years. For long-term performance of prosthetic restorations, ceramic materials must prove chemical stability, strength and reliability in oral environment conditions [1]. Dental ceramics are fabricated with different technologies, varying in their internal structure and composition [2]. The production of multiphase material (hybrid) intends to combine the best mechanical and esthetical characteristics of resin composites and ceramics [3].

Glass-ceramic is composed mainly of silicon dioxide (silica, silica with fillers—usually crystalline fillers, such as leucite or lithium disilicate). A glassy matrix results after firing treatment, and a subsequent heating process is needed to obtain internal crystals [4]. The crystalline phases are the result of the crystallization stage, in which crystal growth and nucleation are produced [1]. Ceramics vary in proportions of glass matrixes and crystalline phases ensuring the balance between mechanical properties and aesthetic aspect [5,6]. Adjusting the volume of those phases, the resistance, dimensional stability, flexural, hardness, strength, fracture toughness and optical effects can be modified and improved [7]. 

Chairside CAD/CAM ceramics can be classified in three categories: silicate ceramics, oxide ceramics and resin matrix ceramics [8]. Oxide ceramics, such as zirconia, have highly mechanical resistance and fracture toughness and glass ceramics are characterized by better aesthetic and translucency, while resin matrix ceramics are known for their improved flexibility, reduced brittleness and rigidity, with better machinability than conventional ceramics [9,10]. Leucite-reinforced ceramic (IPS Empress CAD, Ivoclar Vivadent) is among the first materials comprising a crystalline phase of ceramic, which limits or prevents the amplification of the fracture line. It contains leucite crystals that are uniformly dispersed into the glassy matrix [4]. Lithium disilicate ceramic (IPS e.max CAD, Ivoclar Vivadent) has a crystalline phase integrated into the glassy matrix. This material is partially crystallized, so the CAD/CAM blocks can easily be milled to fabricate the prosthetic restoration, followed by final crystallization [2,11]. Polymer infiltrated ceramic network is a hybrid material consisting of an interpenetrating structure with an inorganic matrix of feldspathic ceramic infiltrated by capillary action with a polymer phase. The combination of the crystalline matrix and the polymeric material results in a decrease in crack propagation [12,13]. 

Ceramic materials and hybrid ceramics are categories of dental materials that have in common the vitreous component, an important factor in adhesive cementation. Adhesive cementation requires both micromechanical and chemical retention [8]. Before applying resin cements, certain pre-treatment procedures are necessary to follow up to obtain micromechanical retention [14,15]. The most widely used surface conditioner of the glass-ceramic is hydrofluoric (HF) acid [16]. Sandblasting with alumina particles is an abrasive method that produces surface roughness and generates the mechanical interlocking space at the interior of restorative materials, which is not always recommended [8,17]. Hydrofluoric acid treatment dissolves the glass matrix from the surface of the ceramic material, displaying the crystals and providing a micromechanically retentive relief in a three-dimensional porous surface, increasing the contact area of the luting cements [18]. Polycrystalline ceramics are not altered by the etchant, and hydrofluoric acid does not modify the intaglio aspect of alumina and zirconia restorations [19]. The acid treatment effects depend on the concentration of the conditioner, the etching time and the ceramic material that is being treated [18]. The durability of ceramic prosthesis depends not only on the mechanical properties and chemical composition of the restoration materials, but also on adhesion, which plays an important role in the endurance of indirect restorations [20]. Adhesive cementation enhances the fracture resistance and improves bond strength of CAD/CAM restorations, preventing the risk of failure [21,22]. Microleakage is also considerably reduced through the adhesive strong bonding, and occlusal forces are more evenly distributed in the resin cement layer, ensuring marginal integrity [3,20,22].

The objective of this study was to evaluate the influence of hydrofluoric acid (HF) and the conditioning time on the micro-shear bond strength (µSBS) of dual cure resin cement to glass-ceramic materials, such as lithium disilicate ceramic (IPS e.max CAD, Ivoclar Vivadent) (EX) and leucite-reinforced ceramic (IPS Empress CAD, Ivoclar Vivadent) (EP), and also a hybrid ceramic (Vita Enamic, Vita Zahnfabrik) (VE).

## 2. Materials and Methods

### 2.1. Specimen Preparation

In the present study, three different CAD/CAM restorative materials (IPS e.max CAD, Empress CAD and Vita Enamic) were tested. The manufacturers and compositions of the materials used in the present study are presented in Table 1. 

Eighteen sections with dimensions of 6 mm × 3 mm × 1 mm from each CAD/CAM block were cut using a slow-speed Buehler IsoMet 1000 precision sectioning saw (BUEHLER, Esslingen, Germany). 

The specimens were randomly divided into 3 groups, each containing 6 samples/test group. Then, 9.5% HF gel (Yellow Porcelain Etch, Cerkamed) was applied on the samples: group 1 for 30 s, group 2 for 60 s, group 3 for 90 s. The etchant was evenly spread for 20 s over the surface of the glass-ceramic and hybrid ceramic samples using a micro-brush. The surface was cleaned with a strong jet of air/water spray for 20 s and dried with oil-free air for 10 s.

A universal adhesive (Scotchbond Universal adhesive (3M ESPE), which also acts similarly to a silane coupling agent, was applied and rubbed onto all pre-treated ceramic surfaces for 20 s using a micro-brush, followed by air thinning for 10 s. The adhesive was cured for 40 s with an LED curing light (DTE LUX-E Plus Curing Light, Woodpecker, power intensity 1000 mW/cm^2^).

A 5 mm long, transparent plastic tube with an internal diameter of 4 mm was filled with adhesive resin cement (RelyX Ultimate 3M ESPE, St. Paul, MN, USA), (Figure 1a).

The tube filled with resin cement was placed on the centre of the ceramic surface and light cured (DTE LUX-E Plus Curing Light, Woodpecker, Guilin, Guangxi, China) for 100 s (20 s on each side). When the adhesive was completely cured (light and chemical), the tube was removed. 

A custom-made silicone cylinder mould with dimensions of 3 cm × 4 cm was filled with acrylic resin (Superacryl Plus, SpofaDental, Jičín, Czech Republic), and the ceramic-adhesive assembly was embedded in the resin, (Figure 1b).

All the specimens were stored in distilled water for several days until the bond strength tests were performed.

### 2.2. Micro-Shear Bond Strength Test

The specimens were subjected to a micro-shear bond strength test (µSBS) using a universal testing machine (Zwick Proline Z005, Ulm, Germany), with a maximum force of 5 kN at the ambient temperature. The shear tests were performed on a special device designed and built to determine the shearing strength. The shearing device was installed on the universal testing machine. The specimens were fixed in a cylindrical appliance. A semi-circular shaped indenter was applied to the load at a crosshead speed of 0.5 mm per minute until failure occurred. From the force displacement curves obtained on several specimens under the shearing tests, the micro-shear bond strength (µSBS) was calculated by dividing the maximum load at failure (N) with the surface area of the cement cylinder (mm^2^).

### 2.3. Failure Mode

The fractured surfaces were microscopically evaluated at ×10 and ×25 magnifications (Olympus SZX7, Tokyo, Japan) to determine the failure modes of specimens, and were classified as: adhesive failure between resin cement and ceramic, cohesive failure within ceramic or cement and mixed failure.

### 2.4. SEM Images

Nine samples of etched substrates (one for each material in each group) were observed by a scanning electron microscope (SEM), (FEI Inspect S, Hillsboro, OR, USA) at different magnifications (50×; 1000×; 5000×). 

### 2.5. Statistical Analysis

Statistical analysis was conducted using IBM SPSS Statistics software to determine the effects of 9.5% HF acid on the micro-shear bond strengths of resin cement on ceramics with respect to etching time. Exploratory data analyses with the Shapiro–Wilk test and Levene test were performed before applying the one-way analysis of variance. One-way ANOVA, followed by the Bonferroni post hoc test with a 95% confidence interval, was used to determine whether distinct etching periods showed significant µSBS mean differences. While applying a one-way analysis of variance for EP, the F-test was used, and for EX and VE materials, the Welch’s test was used.

## 3. Results 

### 3.1. Micro-Shear Bond Strength Test

Mean micro-shear bond strength values (MPa) and standard deviations of the tested materials are shown in Table 2, Table 3 and Table 4.

Displacement curves for some representative specimens of each material are presented in Figure 2, and considering the value of the maximum force, the shear bond strength was determined. For EX, the 60 s etching time had the maximum shear strength (the average was 12.73 MPa), which was not much greater than the shorter etching time (30 s) where the shear strength was 10.6 MPa. At an etching time of 90 s, the shear strength was maximum for EP, showing a good adhesion between the material and the adhesive. The 90 s etching time for VE presented considerably higher shear strength values than the etching times of 30 s and 60 s.

### 3.2. Statistical Analysis

The application of the one-way ANOVA test was preceded by an exploratory data analysis, during which the independent and dependent variables were checked for adequacy. Additionally, verification of whether the dependent variable was approximately normally distributed for each category of the independent variable was conducted using a Shapiro–Wilk test for each group. A check was also conducted to determine whether the dependent variable presented significant outliers in any group. The homogeneity of variances was verified during one-way ANOVA analysis using the Levene statistic test.

After applying a one-way ANOVA test on each data set corresponding to each material, it was concluded that the categorical variable, etching time, had a significant effect on bond strength values (*p* < 0.05) for EP. To determine which differences are statistically significant, the multiple-comparison Bonferroni test was performed. As can be seen, there were statistically significant differences between 60 s and 30 s and between 60 s and 90 s, and there was no statistically significant difference between 30 s and 90 s.

In the case of EX and VE, the Welch’s *p*-value was used to replace the regular ANOVA *p*-value, because the Levene test failed. Hydrofluoric acid with a 9.5% application had no significant effect on bond strength of ceramics for EX (*p* = 0. 809) and VE (*p* = 0.107).

### 3.3. Failure Mode

The failure modes of fractured surfaces were classified as: adhesive failure between resin cement and ceramic, cohesive failure within ceramic or cement and mixed failure. The main failure modes of EX were adhesive failures in all groups. Failure mode examination of EP showed cohesive fractures predominantly within the ceramics. VE presented 51% of the fractures as cohesive failures within ceramic. 

### 3.4. SEM Images

The etched surfaces under the SEM examination for EX (Figure 3a–c) appeared to be needle-shaped, revealing the crystals’ size, form and orientation. The images did not expose significant differences among the three groups of EX, but the depth of the pattern increased with the etching time. SEM images for EP (Figure 3d–f) and VE (Figure 3g–i) showed the glassy matrix removal and exposed the substrate morphology.

## 4. Discussion

The present study assessed the influence of different etching times on the shear bond strength of dual-cure resin cement to leucite-reinforced ceramic (EX), lithium disilicate glass ceramic (EP) and polymer infiltrated ceramic network (VE).

Mechanical tests on dental materials were conducted to understand the performance and durability of bonded restorations, simulating a vast diversity of physical and clinical variations in a short period of time and correlating the obtain results with outcomes collected by other researchers [16,23]. The methods used in vitro to measure the bond strength between the resin luting agents and the ceramic materials are shear and tensile, respectively micro-shear and micro-tensile tests [24]. In the literature, it is conceded that these tests are not universally accepted as testing methods, and each of them have their own advantages and limitations [25]. These tests are recommended for brittle materials, such as glass ceramics [26]. Micro-shear bond strength tests evaluate the adherence of luting materials and the mechanical behaviour of dental ceramics under a load that induces uniform stress concentration at the adhesive interfaces to determine the weakest area between the bonded surfaces [27]. In their studies, Farag, Maño and Sarahneh observed that reducing the size of bonded interfaces decreases the frequency of the defects and undesirable variations in the specimens and obtained lesser cohesive failures [28,29,30]. The stress distribution in the tested samples may be affected by many variables related to the fabrication of specimens, such as the type of cement, the application and the thickness of the cement layer, and also by the testing parameters, such as the distance from the testing device to the cement cylinder or the angle of force application [30]. A disadvantage of sample fabrication is the absence of the compression on resin cement while it is set on ceramic material, which could create a hydraulic force causing the penetration of the cement in all the micropores [14]. In this study, specimens with reduced adhesive areas, for example, with areas 4 mm in diameter, were used to diminish uneven stress distribution in the micro-shear test. 

A good adhesion of the restoration materials to the tooth surface is one of the most important factors for the fixed dental prosthesis’s longevity and endurance [31]. Resin-based cements were introduced onto the market when all-ceramic restorations began to be widely utilized. These cements are essential when it comes to bonding indirect ceramic restorations and CAD/CAM materials. They need to be used with a bonding agent to increase bond strength. Resin cements are typically dual-cured to ensure adequate polymerization, and usually have a higher strength than self-adhesive cements, and even than conventional cements [32]. Adhesive cements show reduced dissolution and microleakage, increased fracture resistance and high compressive and tensile strength, and depending on the type of cement, may be easy to manipulate, or may require more steps before application [31]. Adhesive systems are categorized as total etch, self-etch and self-adhesive, and their retention is a combination of three mechanisms—chemical, mechanical and micromechanical [31,33]. To achieve an adequate adhesion and chemical and micromechanical retention, a surface pre-treatment of ceramic restorations and dentine is required before the cementation [34,35].

Micromechanical interlocking of ceramic restorations is attained by using HF acid etching on its intaglio surface, but for the chemical bonding, a silane coupling agent must be applied [9,15]. The adhesive used in this study already contains the silane. The purpose of this type of adhesive is to provide bonding to glass ceramics or resin composites without additional priming procedures.

When applied onto the ceramic surface, HF acid reacts with the silica matrix, dissolving and removing the surface layer of the glassy matrix containing silica, silicates and leucite crystals [36]. Each ceramic material displays a different, individual etching design in accordance with the ceramics’ composition and their crystalline and vitreous phase distributions [37], as seen in Figure 3. HF acid leaves behind a surface prepared for penetration and diffusion of the luting material, increasing the bond strength between restoration and adhesive with an expanded geometrical pattern in the ceramic structure, and also enhancing the wettability and surface energy of the substrate [38,39]. 

In the present study, the SEM images showed that the glass matrix dissolved faster than crystals. The dissolution of the vitreous phase was more obvious for EP and VE as the etching time gradually increased. For VE, the polymeric substrate became clearer when etching time was prolonged to 90 s. It was clearly observed that HF enhanced the asperities of the material’s surface and changed the ceramic structure with the reduction in the feldspathic network. The polymer component in VE was not altered by enchant. EP appeared to grow deeper gaps between crystals. The loss of glassy phase enlarged the pores with the increasing in the etching time. The HF acid action on EX exposed needle-like crystals oriented in all directions. 

It was observed that with the increase in of the crystallization rate of CAD/CAM dental restoration materials, the solubility level decreased [40]. Additionally, there were substantial differences in the proportion of chemical components among the products from diverse manufacturers, which led to various etching-time procedures [41]. The recommended conditioning practice comprised etching with 5% to 10% hydrofluoric acid (HF), with etching time varying from 15 s to 90 s [42,43]. The manufacturer’s etching recommendations for lithium disilicate ceramic was 20 s with 5% HF acid, while for feldspathic ceramic, leucite-reinforced feldspathic ceramic and polymer infiltrated ceramic network required 60 s with 9%–10% HF [10]. In their review, Nogueira et al. reported that the studies regarding HF etching applications and protocols have contradictory results, and optimal treatment has not yet been defined [16]. 

Raising the etching time had no significant influence on the bond strength of ceramics for EX (*p* = 0.809) and VE (*p* = 0.107) although the conditioning time improved the bond strength for all tested CAD/CAM materials. Etching the surface of the lithium disilicate ceramic with 9.5% HF for 60 s produced the highest µSBS.

The lower shear bond strength results were recorded in the case of VE, because the material’s structure is based on polymer content and weak interatomic bonds. EX showed a greater ceramic density compared to EP, due to its high crystalline content. Regarding EP, the material density decreased with the increase in etching time. 

In the literature, supplementing the etching time and the concentration of HF for lithium disilicate have led to inverse results. Souza et al. noticed that HF etchant may induce a higher or incomplete dissolution of the glassy phase in the ceramic materials due to exposure time or acid concentration [44]. Some studies tested different HF etching times and found that increasing the demineralization period above 30 s led to a stronger adhesion between feldspathic ceramic and a resin cement [37,43]. Other authors suggested that etching time under 60 s showed minor surface alteration, but increasing the dissolution time, a deeper pattern was obtained [45]. 

El-Damanhoury and Gaintantzopoulou reported that over-etching lithium disilicate doesn’t improve the material’s bond strength as much as the chemical mechanism in the resin cement. They also noticed that HF dissolved the glassy phase and the polymer in the polymer-infiltrated ceramic-network reducing the retentive design, which might be considered a negative effect in the adhesion protocol [20].

There were contradictory reports regarding HF acid concentrations of 5% and 10% in SBS mean values considering CAD/CAM materials such as lithium disilicate, feldspar, leucite or hybrid ceramics, but most of the studies agreed that HF acid modifies the ceramic surface geometry and morphology [43]. 

Kalavacharla et al. obtained greater shear bond strength values for lithium disilicate specimens etched for 60 s with 9.5% HF acid than for those etched for 20 s with 5% HF acid, but not treated with silane. They explained that the 9.5% HF acid produced a deeper dissolution of the material, and the bond strength relied on the micromechanical retention. The authors were also concerned that over-etching the ceramic might reduce the strength of the material [46]. Other studies have also raised the aspect of the weakening effect that HF acid produces, regardless of the action time or concentration [41,45].

Some studies confirmed that the HF etching time and concentration for ceramic materials should be increased to over 60 s to produce a sufficient material displacement for adequate retention and bond strength. Superficial roughness was obtained for etching times of 20 s and 60 s, which might not provide an appropriate mechanical interlocking effect [37,41,43,45,46]. 

Failure mode examination of EP showed cohesive fractures predominantly within the ceramics, which revealed that the adhesion between the resin cement and base material was stronger compared to the shear resistance of the ceramic. The predominant failure modes of EX were adhesive failures in all groups. This does not necessarily represent a weak bond between adhesive and base material, but a durable ceramic. However, low bond strength values correspond to adhesive failures. VE presented 51% of the fractures as cohesive failures within the ceramic. The results revealed that the surface treatment enlarged the asperities and produced deeper grooves by removing the silica content of the glassy matrix. This preferential disintegration of the vitreous phases of ceramic materials creates a micromechanically retentive environment, regardless of the tested material, amplifying the contact surface between ceramic and adhesive, although there is a correlation between the failure mode and the CAD-CAM material type. The results of our study confirmed that an increase in the HF acid etching time induced a greater amount of ceramic dissolution, which resulted in a higher-strength bond to resin cement. However, prolonged etching does not significantly increase the bond strength. 

## 5. Conclusions

The increase in the HF acid concentration and duration induced a greater amount of ceramic dissolution, both on the surface and in the depth of the material, which resulted in a higher-strength bond to resin cement.

For leucite-reinforced ceramic and hybrid ceramic, an etching time of 90 s showed good µSBS results; therefore, a prolonged conditioning time will not negatively affect these CAD/CAM materials.

In the case of lithium disilicate, superior CAD/CAM ceramic results were obtained at 60 s etching time with a concentration of 9.5% HF acid. There was no great difference between the 30 s and 60 s values, which led to the conclusion that for lithium disilicate, it is appropriate to use either a higher acid concentration with manufacturer-recommended time or a higher etching time with manufacturer-recommended acid concentration.

Further investigations under clinical conditions are necessary regarding the bond strengths of CAD/CAM materials and resin cements, and also regarding surface pretreatment. 

## Figures and Tables

**Figure 1 materials-15-07071-f001:**
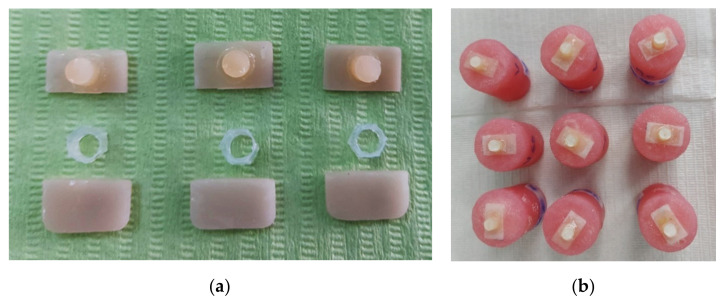
(**a**) Experimental design featuring a sample preparation where the plastic tube was filled with resin cement.; (**b**) samples prepared for micro-shear bond strength tests.

**Figure 2 materials-15-07071-f002:**
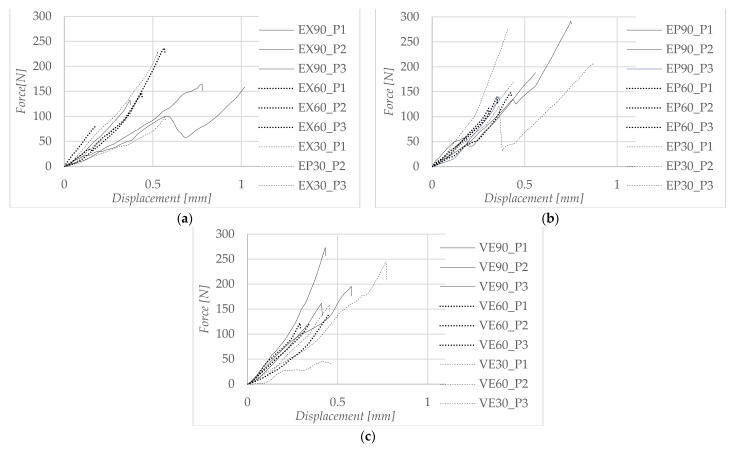
Force–displacement curves for some representative specimens. (**a**) Sample of EX: 9.5% HF for 30 s, 60 s and 90 s; (**b**) sample of EP: 9.5% HF for 30 s, 60 s and 90 s; (**c**) sample of VE: 9.5% HF for 30 s, 60 s and 90 s.

**Figure 3 materials-15-07071-f003:**
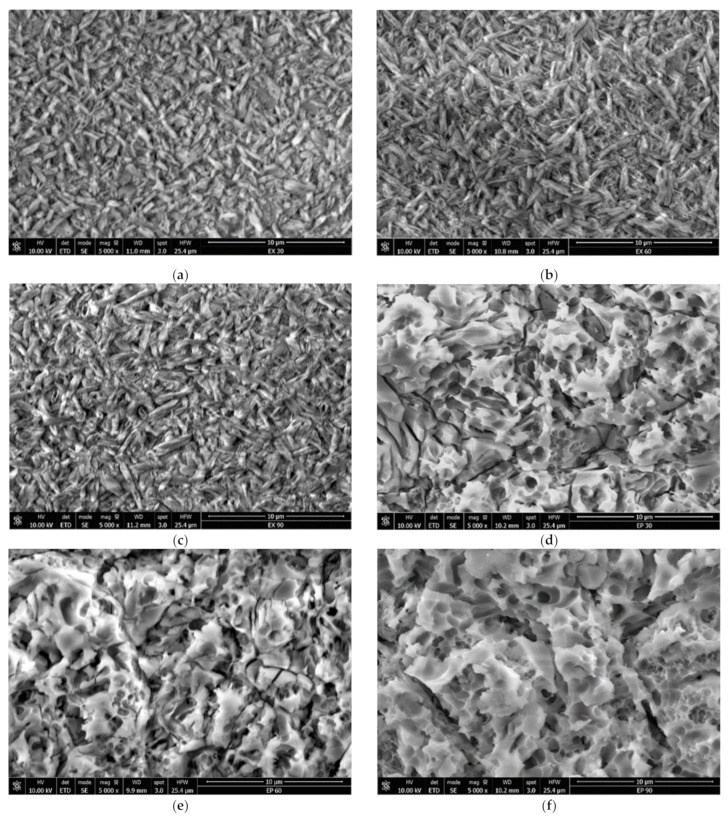
Scanning electron images of the representative etched sample. Ceramic crystals were visible following the removal of the glassy phase (×5000). (**a**) Sample of EX: 9.5% HF for 30 s; (**b**) sample of EX: 9.5% HF for 60 s; (**c**) sample of EX: 9.5% HF for 90 s; (**d**) sample of EP: 9.5% HF for 30 s; (**e**) sample of EP: 9.5% HF for 60 s; (**f**) sample of EP: 9.5% HF for 90 s; (**g**) sample of VE: 9.5% HF for 30 s; (**h**) sample of VE: 9.5% HF for 60 s, and (**i**) sample of VE: 9.5% HF for 90 s.

**Table 1 materials-15-07071-t001:** Materials used in the study.

Material	Type	Manufacturer	Composition
Empress CAD	Leucite-reinforced glass ceramic CAD/CAM block	Ivoclar-Vivadent Schaan Liechtenstein	Leucite-reinforced glass-ceramic (SiO_2_, Al_2_O_3_, K_2_O, Na_2_O and pigments)
IPS e.max CAD	Lithium disilicate glass ceramic CAD/CAM block	Ivoclar-Vivadent Schaan Liechtenstein	Glass ceramic with lithium disilicate fillers (SiO_2_, Li_2_O, K_2_O, MgO, Al_2_O_3_, P_2_O_5_, other oxides and pigments)
Vita Enamic	Polymer infiltrated CAD/CAM block	Vita Zahnfabrik, Bad Säckingen, Germany	86 wt% feldspar ceramic, 14 wt% polymer (UDMA, TEGDMA) SiO_2_, Al_2_O_3_, Na_2_O, K_2_O, B_2_O_3_, CaO, TiO_2_, TEG-DMA, UDMA
Yellow Porcelain Etch	Ceramic etching gel	PPH Cerkamed Wojciech Pawlowski Stalowa Wola, Poland	Hydrofluoric acid 9.5%
Single Bond Universal	Universal adhesive	3M ESPE Dental Products, St. Paul, MN, USA	MDP phosphate monomer. Dimethacrylate resins. HEMA. Vitrebond. Copolymer. Filler. Ethanol. Water. Initiators. Silane
RelyX Ultimate Adhesive	Dual-cure resin cement	3M ESPE Dental Products, St. Paul, MN, USA	10-Methacryloxydecyl dihydrogenphosphate (MDP) Dimethacrylate resins. HEMA. Vitrebond. Copolymer. Filler. Ethanol. Water. Initiators. Silane

**Table 2 materials-15-07071-t002:** Descriptive statistics results for EP.

Descriptive—EP								
Time	N	Mean	Std. Deviation	Std. Error	95% Confidence Interval for Mean		Minimum	Maximum
					Lower Bound	Upper Bound		
30	6	17.42467	3.842316	1.568619	13.39240	21.45693	12.901	22.343
60	6	11.06450	1.146762	0.468164	9.86105	12.26795	9.394	12.376
90	6	18.10983	4.733010	1.932243	13.14284	23.07682	11.286	23.792
Total	18	15.53300	4.687373	1.104824	13.20202	17.86398	9.394	23.792

**Table 3 materials-15-07071-t003:** Descriptive statistics results for EX.

Descriptive—EX								
Time	N	Mean	Std. Deviation	Std. Error	95% Confidence Interval for Mean		Minimum	Maximum
					Lower Bound	Upper Bound		
30	6	10.60383	5.237264	2.138104	5.10766	16.10000	5.933	18.260
60	6	12.73050	5.419664	2.212569	7.04291	18.41809	6.489	19.607
90	6	11.68317	0.712398	0.290835	10.93555	12.43078	10.586	12.394
Total	18	11.67250	4.201647	0.990338	9.58307	13.76193	5.933	19.607

**Table 4 materials-15-07071-t004:** Descriptive statistics results for VE.

Descriptive—VE								
Time	N	Mean	Std. Deviation	Std. Error	95% Confidence Interval for Mean		Minimum	Maximum
					Lower Bound	Upper Bound		
30	6	9.97600	4.406027	1.798753	5.35216	14.59984	2.706	14.284
60	6	10.09017	1.260952	0.514781	8.76688	11.41345	8.703	11.895
90	6	15.13233	4.657223	1.901303	10.24488	20.01979	10.387	22.871
Total	18	11.73283	4.321713	1.018638	9.58370	13.88197	2.706	22.871

## Data Availability

Not applicable.

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
