# Peer review of "Effect of Different Etching Times with Hydrofluoric Acid on the Bond Strength of CAD/CAM Ceramic Material"

_materials, 2022, doi:10.3390/ma15207071_

Round 1

Reviewer 1 Report

- The introduction section sounds rare; introducing various materials without any correlation makes it difficult to understand the authors' main idea; review this section and rewrite it. The authors should state the novelty of their research regardless state of the art in the field of study.

- The Fig. 1 caption is out of position; please correct this mistake.

- The authors must avoid describing their results in Section 2; this is the case with the SEM micrographs (Fig. 3). These results must be described in Section 3, and select the relevant results to include in the final Fig. 3; there are many pictures in Fig 3.

- There is no correlation between the sentences in the discussion section it is difficult to analyse the new research results. I suggest the authors read each sentence carefully and compare them to provide solid sentences describing the further information from their research.

- The conclusion must be checked. Currently, these only compile your results; therefore, the novelty of your work is not apparent to the potential readers; please check this section and rewrite it.

Author Response

Dear reviewer,

Thank you for your time to follow our manuscript. Also, we thank you for your recommendations to improve the manuscript. Please, see below the answers to your recommendations/observations. We have carefully followed your suggestions, then we have processed all of them. Please find below the answer to each comment received from your side.

  1. The introduction section sounds rare; introducing various materials without any correlation makes it difficult to understand the authors' main idea; review this section and rewrite it. The authors should state the novelty of their research regardless state of the art in the field of study.

A1: The introduction section was revised.

  1. The Fig. 1 caption is out of position; please correct this mistake.

A2: The mistake was corrected.

  1. The authors must avoid describing their results in Section 2; this is the case with the SEM micrographs (Fig. 3). These results must be described in Section 3, and select the relevant results to include in the final Fig. 3; there are many pictures in Fig 3.

A3: The results section was revised and reorganized.

  1. There is no correlation between the sentences in the discussion section it is difficult to analyze the new research results. I suggest the authors read each sentence carefully and compare them to provide solid sentences describing the further information from their research.

A4: The discussion section was revised and reorganized.

  1. The conclusion must be checked. Currently, these only compile your results; therefore, the novelty of your work is not apparent to the potential readers; please check this section and rewrite it.

A5: The conclusion section was rewritten.

Best regards,

The Authors

Reviewer 2 Report

Dear Authors, 

My sincere appreciation for the efforts involved. However, I would like you to go through the comments for not recommending the article for publication.

Author Response

Dear reviewer,

Thank you for your time to follow our manuscript. Also, we thank you for your recommendations to improve the manuscript. Please, see below the answers to your recommendations/observations. We have carefully followed your suggestions, then we have processed all of them. We introduced a lot of new details in the manuscript taking into account your essential comments. Please find attached the answer to each comment received from your side.

Reviewer 3 Report

Dear authors,

the article covers a very interesting topic.

Nevertheless, I suggest some changes in order to improve the overall quality of the manuscript for the readers.

First of all English proofreading is mandatory. Especially for the discussion.

Line 29: 

The authors wrote:

”Surface roughness was examined” maybe the authors intended “Etched surface was examined” ?

Lines 35-6:

“etching” and “etching time” could be added to the Keyword list.

Lines 94-95:

The authors mention 1) improved fracture resistance and 2) improved bond strength.

The cited reference [25] does not mention anything about fracture resistance. The authors could consider in also adding the following reference (that is also related to Lithium disilicate - one of the investigated materials):

Baldi A, Comba A, Michelotto Tempesta R, Carossa M, Pereira GKR, Valandro LF, Paolone G, Vichi A, Goracci C, Scotti N. External Marginal Gap Variation and Residual Fracture Resistance of Composite and Lithium-Silicate CAD/CAM Overlays after Cyclic Fatigue over Endodontically-Treated Molars. Polymers (Basel). 2021 Sep 4;13(17):3002. doi: 10.3390/polym13173002. PMID: 34503042; PMCID: PMC8434150.

Line 98:

The authors could also cite the above-mentioned reference that makes reference to marginal integrity.

Line 125:

The authors could add the power intensity of the curing light: mW2

Lines 152-4:

Please specify how the fracture’s surfaces were evaluated. With opticla microscope? With the SEM?

Lines 292 and 295:

The authors could remove new paragraph break/new line and make a whole paragraph.

Line 304 and 313 and 326 and 333 and 340 and 342 and 347 : The authors could remove new paragraph break/new line

Line 319:

The authors wrote:

“HF acid is a colourless liquid, corrosive to metals and tissue. The liquid or its vapours can cause severe, painful burns [48].”

Commercially HF acid is always colored (yellow or red). Please add this information. “Colourless” may be confusing for the reader

LINE 348:

The authors wrote “El-Damanhoury find out that not over etching” 

Please change the verb in the past mode. Maybe use: “reported”

Line 357:

The authors wrote:

“Kalavacharla obtained greater shear bond strength”

The authors should change to:

Kalavacharla et al.

while there are more authors.

Line 359: same as the point before “He explained…”

Line 361: same as the point before “The author also is”

Discussion (around line 359):

The authors could add a paragraph related to silane usage. Used as a separate liquid/step or inside a universal adhesive as the authors performed. Why the authors have decided to use the adhesive way and possible differences in performance and bond strength.

Author Response

Dear reviewer,

Thank you for your comments and your appreciation. We have examined your recommendations, then we have answered to them. Please find attached the answer to each comment received from your side.

Round 2

Reviewer 2 Report

Dear Authors,

I am satisfied with the amendments.